# Experimental and Numerical Investigations of Thin-Walled Stringer-Stiffened Panels Welded with RFSSW Technology under Uniaxial Compression

**DOI:** 10.3390/ma12111785

**Published:** 2019-06-01

**Authors:** Andrzej Kubit, Tomasz Trzepiecinski, Łukasz Święch, Koen Faes, Jan Slota

**Affiliations:** 1Department of Manufacturing and Production Engineering, Rzeszow University of Technology, al. Powst. Warszawy 8, 35-959 Rzeszów, Poland; 2Department of Materials Forming and Processing, Rzeszow University of Technology, al. Powst. Warszawy 8, 35-959 Rzeszów, Poland; tomtrz@prz.edu.pl; 3Department of Aircraft and Aircraft Engines, Rzeszow University of Technology, al. Powst. Warszawy 8, 35-959 Rzeszów, Poland; lukasz.swiech@prz.edu.pl; 4Belgian Welding Institute, Technologiepark Zwijnaarde 935, B-9052 Ghent, Belgium; koen.faes@bil-ibs.be; 5Institute of Technology and Material Engineering, Faculty of Mechanical Engineering, Technical University of Košice, Mäsiarska 74, 040 01 Košice, Slovakia; jan.slota@tuke.sk

**Keywords:** finite element modeling, friction stir spot welding, thin-walled structure, post-buckling analysis

## Abstract

Many aircraft structures are thin walled and stringer-stiffened, and therefore, prone to a loss of stability. This paper deals with accurate and validated stability analysis of the model of aircraft skin under compressive loading. Both experimental and numerical analyzes are conducted. Two different methods of joining panel elements are considered. In the first case, the panel is fabricated using rivets. In the second variant, the refill friction stir spot welding technique is used. Both types of panels are loaded in axial compression in a uniaxial tensile testing machine. The geometrically and physically nonlinear finite element analyzes of the panels were carried out in ABAQUS/Standard. The Digital Image Correlation measurement system ARAMIS has been utilized to monitor the buckling behavior and failure mode in the skin-stringer interface of the stiffened panels. The results of experiments and the digital image correlation system are presented and compared to the numerical simulations.

## 1. Introduction

Thin-walled stringer-stiffened structures are widely used in many areas of engineering, both in the construction industry, and manufacturing mechanical constructions. As far as the technologies for producing the thin-walled stiffened structures used in the aircraft industry are concerned, in recent years there has been a great interest in new technologies for joining the skin to the stringers. In most aircraft, skins of riveted panels are made of aluminum alloy clad sheets [1]. Riveted joints became the main technique of forming connections with welding. The technique of joining thin-walled aircraft panels by riveting has been the traditional method that has been used for decades, and alternative technologies are now being developed [2,3].

Friction Stir Welding (FSW), an alternative joining process to traditional rivet fastening in primary aircraft structures, has the potential to lower manufacturing costs and structural weight [4]. The new welding technologies for joining thin-walled structural elements include linear and spot variants of friction welding. 

FSW assures lower labor-consumption and better properties of the resulting joints when compared to riveted structures. Many papers have been devoted to strength analysis of FSW fabricated joints. A wide variety of materials, thicknesses and joint configurations have been investigated using this process [5]. Sepe et al. [6] conducted experimental tests and numerical computations to evaluate the fracture arrest capability of welded stiffeners on flat panels subjected to fatigue loads. They found that the Dual Boundary Element Method (DBEM) procedure simulates crack growth more efficiently and accurately than the Finite Element Method (FEM). Citarella et al. [7] analyzed the influence of the FSW process-induced residual stresses on the fatigue behavior of welded butt joints in 2024-T3 aluminum alloy sheets. The proposed FEM-DBEM approach successfully predicted crack growth in the presence of residual stresses induced by the manufacturing process. Sekhar et al. [8] and Subhashini et al. [9] conducted optimization studies of FSW technology by selecting the proper parameters for the process of joining different materials. Friction stir welding (FSW) is used by Spirit Aero Systems in the production of nose barrier beams for the Boeing 747 freighter. Furthermore, the solid-state welding technique was also adopted for joining fuel tanks on satellite launch vehicles. The Welding Institute (Cambridge, UK) is assisting Embraer in introducing FSW into the manufacture of forward fuselage panels on the Legacy 450 and 500 aircraft [10].

Refill friction stir spot welding (RFSSW) was developed by Helmholtz-Zentrum Geesthacht in Germany [2] in order to achieve a higher weld strength compared to conventional friction stir spot welding. The RFSSW solid-state technique has the most outstanding advantage of fabricating spot welds without a keyhole. In the case of the RFSSW method, many scientific papers [11,12,13] have been written about joining the 2xxx and 7xxx series aluminum alloy sheets used for primary airframe structures. Kubit et al. [13] investigated the effect of structural defects on the fatigue strength of Alclad 7075-T6 aluminum alloy RFSSW joints. This paper also discussed the effect of the weld defects associated with material flow on the failure mechanism of single lap joints. Shen et al. [11] investigated the microstructure and mechanical properties of 7075-T6 aluminum alloy joints joined by RFSSW. It was found that an optimal selection of the welding parameters played an important role in determining the strength of the joint, and the main feature affecting the mechanical properties of the joint is the Alclad between the upper and lower sheets. Reimann et al. [14] studied the thermal cycle and the evolution of microstructural features in terms of the mechanical properties of 7075-T651 aluminum alloy joints.

The semi-monocoque system is the most frequently used load-bearing structure for high-performance aircraft. This system uses a substructure to which the airplane’s skin is attached. The stringer-stiffened panels can carry a considerable load after buckling in the elastic range. The efficiency of the stiffened panel structure is influenced by the interaction of materials, geometric design and the assembly (joining) technology. Stability analyzes of thin-walled structures were investigated for many years by analytical methods [15,16,17,18]. A problem of the calibration of the computer model with experimental works in the area of metal structures have been considered before by Szafran et al. [19]. The stable postbuckling response of stiffened panels is a beneficial characteristic [20]. The ultimate strength of stiffened steel panels has been the subject of many investigations while the literature on stiffened aluminum-alloy panels is more limited. Khedmati et al. [21] addressed a numerical investigation concerning postbuckling behavior and the ultimate strength of aluminum panels using the FE-based ANSYS program. They found that the stress-strain curves derived under different values of loading have a significant effect on the postbuckling response of stiffened panels. Wadee and Farsi [22] analyzed an analytical model of the buckling behavior of a thin-walled stiffened plate subjected to axial compression.

Residual stresses, as the consequence of joints fabrication, influence the mechanical properties and fatigue strength of joints and components. Expansion of the rivet shank during installation produces an interference that results in a residual stress field around the rivet hole. The interference fit provided by solid rivets introduces a residual stress field beneficial to the fatigue life of riveted joints [23]. Although FSW produces low-distortion welds of high quality, significant levels of residual stresses may arise. 

These residual stresses, due both to mechanical and thermal causes, can influence the service performance of the welded components with respect to corrosion properties and fatigue life [24]. Anastassiou et al. reported that residual stresses in the welding region decrease the fatigue and fracture strength of the welds manufactured by the spot welding process [25]. A considerable amount of compressive residual stress is formed at the thermomechanical affected zone (TMAZ) because of the synergy between the thermal expansion, due to the heat conduction from the stir zone (SZ) and the mechanical compression by the tool. Both the formation of compressive residual stress in the TMAZ and the mitigation of residual stress in the SZ contribute to the improvement of the mechanical properties of the friction stir welded joints [26]. In the case of adhesive joints, the curing process induces a significant amount of thermal residual stress in the adhesive layer of joints. Ghazijahani et al. [27] studied the effect of partial and full-length stiffening of shells in which the stiffeners were attached without welding to avoid the adverse effect of the residual stresses. The epoxy adhesive was employed to connect the stiffeners. This connection method helped the structures have uniform material properties in comparison with the welded or soldered connections, in which a lot of residual stress can affect the buckling behavior of such structures [27,28]. Humfeld and Dillard [29] indicated that residual stresses in an elastic-viscoelastic bimaterial system incrementally shift over time when subjected to thermal cycling. They also found that the increasing tensile residual stresses induced in an adhesive bond subjected by thermal cycling may lead to damage and debonding, thus reducing bond durability.

This paper presents what is intended to be an accurate and validated stability analysis of a model of aircraft skin under axial compression. The investigations consisted in an experimental and finite element-based computations using the ABAQUS/Standard program. Two methods of joining the stringer-stiffened panel elements are used, i.e., riveting and refill friction stir spot welding. A postbuckling analysis of the panels under axial compression was also undertaken using the finite element method and digital image correlation technique.

## 2. Materials and Methods

### 2.1. Material

The panels stiffened with attached stringers were made of Alclad 7075-T6 aluminum alloy with zinc as the primary alloying element. This kind of aluminum alloy is widely used in aircraft constructions. The chemical composition of the main alloying elements (wt %) is as follows: Cu—1.2~2.0, Mg—2.1~2.9, Zn—5.1~6.1, Mn—0.30, Fe—0.50, Si—0.40, Ti—0.30, other impurities 0.15 in total. The thickness of the skin plate was 0.8 mm, and the thickness of the stringer, was 1.6 mm. This configuration corresponds to the joining of a stringer with the skin in an actual aircraft structure. When the panel is in a riveted configuration, rivets made of PA24 aluminum alloy with a diameter of 3 mm are used. PA24 (AlCu2Mg) material is an equivalent of the 2117 aluminum alloy. The yield stress of PA24 material is equal to 202 MPa. The values of the basic mechanical parameters (Table 1) were determined by tensile testing according to the ISO 6892-1:2016 standard [30]. Three samples were tested, and the average values of basic mechanical parameters were determined.

### 2.2. Fabrication of Stringer-Stiffened Panels

Figure 1 presents a view and the dimensions of a panel stiffened by two stringers. The skin and stringers were joined in the configuration of a single line lap joint. Such a structure was treated as the model of a fuselage skin which is connected to stringers using the two main techniques of forming connections in aircraft structures: Riveting and Welding. 

The pitch spacing used between connectors depends on the connector diameter, and was in accordance with recommendations for the aircraft industry.

Refill friction stir spot welding (RFSSW) panels were fabricated with two connector spacings: *s* = 29.5 mm and *s* = 44.25 mm. In the case of riveted joints, single line rivet connections were fabricated with a spacing *s* = 23.5 mm. One panel has been prepared for each configuration. The riveting process was conducted in a universal riveting machine. The refill friction stir spot welding process was conducted using an RPS100 spot welder by Harms & Wende GmbH & Co KG (Hamburg, Germany) at the Belgian Welding Institute (Zwijnaarde—Ghent, Belgium). The RFSSW welding process can be briefly divided into four main stages: Touchdown, plunging, refilling and tool retracting, which were described in detail in the authors’ recent paper [13]. The parameters of the RFSSW were as follows: Spindle rotation speed 2000 rpm, plunge time 0.5 s, stirring dwell time 1.5 s, tool retract time 0.5 s and plunge depth 1.6 mm. The diameter of the welds was 9 mm.

### 2.3. Compression Test

In the compression tests a universal Zwick/Roell Z100 testing machine (ZwickRoell GmbH & Co.KG, Ulm, Germany) was used. The panel was compressed between the two grippers of the testing machine according to the configuration shown in Figure 2. The longitudinal edges of the panels were free, while the stiffened panel was simply supported along the transverse boundaries. The lower and upper parts of the panel were locked in special holders mounted on the machine grippers along a length of 30 mm. During the tests, an axial compressive load was applied with an increase in the load of 500 N·s^−1^.

### 2.4. Digital Image Correlation System

Loading of the panel up to collapse is conducted using the ARAMIS system (GOM GmbH, Braunschweig, Germany, Germany). ARAMIS is a non-contact and material-independent measuring system based on three-dimensional digital image correlation (DIC). DIC allows one to monitor and analyze the development of strain and displacements in time. DIC relies on a contrasting pattern applied onto the panel surface. The pattern has been artificially applied in the form of random, isotropic and highly contrasting spots.

The quality of the pattern influences the accuracy and resolution of the results. Measuring with the use of the ARAMIS system consists of taking a series of photos of the object being analyzed. The interference fringe pattern related to the element shape is recorded by the matrices of two stereo-metrically arranged digital cameras, so that it is possible to determine the spatial position of each visible point. The resulting spatial point cloud is subjected to the so-called triangulation process which results in the creation of an initial 3D digital model [12].

### 2.5. Numerical Modeling

Numerical modeling of the panel compression was carried out in the ABAQUS/Standard (Dassault Systèmes, Vélizy-Villacoublay, France) software. ABAQUS allows one to analyze physical models of real processes putting special emphasis on geometrical non-linearities caused by large deformations, material non-linearities and complex friction conditions. The stringer-stiffened panel was modeled in accordance with the experimental setup (Figure 1). The boundary conditions of the panel were defined in such a way that nodes at the lower edge were completely clamped, while those at the upper edge were restrained at all six DOFs, except for the in-plane motion along the y-coordinate axis (Figure 2a). The panels are axially loaded, imposing a compressive, uniformly distributed loading over the width of the plate. A uniformly distributed load is applied directly to the surfaces of both stringers and skin (Figure 2b).

The skin-stringer models of the stiffened panel were modeled with 4-node reduced integration, doubly curved shell elements called S4R in ABAQUS terminology [31]. S4R is a 4-node, quadrilateral, stress/displacement shell element with reduced integration and a large-strain formulation. These element types permit transverse shear deformation. Five integration points were employed through the thickness direction. The plate model was composed of 8646 elements, while the stringers consisted of 4716 elements. Due to the small mutual displacements of the parts being joined, frictionless contact conditions were assumed. The sheet material was treated as an elastic-plastic material with isotropy governed by the Huber-Mises-Hencky theory on equivalent stress. In addition, an isotropic work hardening described by the Hollomon law is assumed. 

The strain hardening curve corresponds to the results of the uniaxial tensile test. The density of the 7075-T6 aluminum alloy is assumed to be 2780 kg·m^−3^. The total stress in the range of Hook’s law is defined from the total elastic strain as:(1)$$\sigma =D^{e}\cdot \varepsilon^{e}$$
where *σ* is the total stress, *D^e^* is the fourth-order elasticity tensor and *ε^e^* is the total elastic strain.

In the case of linear elasticity the relation between stress and strain is given by
(2)$$\left\{\begin{array}{c} \varepsilon_{11}\\ \varepsilon_{22}\\ \begin{array}{c} \varepsilon_{33}\\ \gamma_{12}\\ \begin{array}{c} \gamma_{13}\\ \gamma_{23} \end{array} \end{array} \end{array}\right\}=\left[\begin{array}{cccccc} \frac{1}{G} & \frac{-\nu }{E} & \frac{-\nu }{E} & 0 & 0 & 0\\ \frac{-\nu }{E} & \frac{1}{G} & \frac{-\nu }{E} & 0 & 0 & 0\\ \frac{-\nu }{E} & \frac{-\nu }{E} & \frac{1}{G} & 0 & 0 & 0\\ 0 & 0 & 0 & \frac{1}{G} & 0 & 0\\ 0 & 0 & 0 & 0 & \frac{1}{G} & 0\\ 0 & 0 & 0 & 0 & 0 & \frac{1}{G} \end{array}\right]\left\{\begin{array}{c} \sigma_{11}\\ \sigma_{22}\\ \begin{array}{c} \sigma_{33}\\ \sigma_{12}\\ \begin{array}{c} \sigma_{13}\\ \sigma_{23} \end{array} \end{array} \end{array}\right\}$$
where: *ε*_ii_ and *γ*_ii_ are the normal and shear strains, respectively; *σ*_ii_ are the stress components; *E* is the Young’s modulus and *G* is the shear modulus expressed in terms of *E* and Poisson’s ratio *ν* as
(3)$$G=\frac{E}{2\left(1+\nu \right)}$$

In the numerical simulation the elastic behavior of panel material was specified by the value of Young’s modulus, *E* = 73,000 MPa, and Poisson’s ratio, *ν* = 0.33.

Implementation of the von Mises yield criterion in the implicit analysis of the finite element-based model must take into account some points. The numerical model should comprise a linear elastic law according to Equation (1). A von Mises yield function can be written in the form:(4)$$\Phi \left(\sigma ,\;\sigma_{y}\right)=\sqrt[{3}]{3J_{2}\left(s\left(\sigma \right)\right)}-\sigma_{y}$$
where $\sigma_{y}$ is the uniaxial yield stress dependent on the accumulated plastic strain ${\bar{\varepsilon }}_{p}$:

Furthermore the numerical analysis should comprise an associative hardening rule, with evolution for the hardening internal variable given by [32]:(5)$$\dot{{\bar{\varepsilon }}_{p}}=\sqrt{\frac{2}{3}}∥\dot{\varepsilon_{p}}∥$$
where $\varepsilon_{p}$ is a plastic strain tensor.

An associative Prandtl-Reuss flow rule is given by the relation:(6)$$\dot{\varepsilon_{p}}=\dot{\gamma }N=\gamma \frac{\partial \Phi }{\partial \sigma }$$
where flow vector *N*:(7)$$N=\frac{\partial \Phi }{\partial \sigma }=\sqrt{\frac{3}{2}}\;\frac{s}{∥s∥}$$
where *s* denotes the stress deviator.

The welded and riveted joints were modeled using mesh-independent point-based fasteners between the plate and stringers. The WELD-type and BEAM Multi-Point Constraints (MPCs) [31] were used to model spot welded and riveted joints, respectively. These types of fasteners can be located anywhere between the parts that are to be connected regardless of the mesh. The diameters of the connectors corresponded to the experimental conditions.

To solve the governing equations by applying the unbalanced forces and computing the corresponding displacements, the Newton-Raphson method is used. To solve the Newton-Raphson’s iterations, the time step was controlled by the ABAQUS automatic incrementation technique.

## 3. Results and Discussion

The experimental analysis of the deformation states during the uniaxial compression test of a plate, stiffened with two stringers in riveted and welded variants, with a spacing of *s* = 29.5 mm, showed a similar, almost linear, increase in the force value (Figure 3). In the case of a welded panel with a spacing between welds of *s* = 44.25 mm, the plate exhibits a lower stiffness, and the force value was lower than in the case of the rest variants.

One of the welded joints in the panel was damaged before reaching the maximum compressive force. This is revealed by the local sudden decrease in the loading force. In the aircraft industry weld and rivet failure is a crucial phenomenon. Wang [33] investigated the load transfer mechanism for different types of riveted joints. He found that a higher squeeze force can cause higher clamping pressure on the panel, and the higher load can be better transferred from one rivet to the adjacent rivet. The presence of a crack in the riveted joint drastically changes the load transfer capability; an uncracked panel can transfer a 60% load, while under the same conditions a cracked panel can only transfer a 40% load [33].

The analysis of deformation states of the panels was carried out on the basis of three-dimensional digital image correlation analysis using the ARAMIS system. For example, four quantitatively measured deformation patterns at characteristic load levels are depicted in Figure 4 and Figure 5. These use images obtained from the DIC system. The deflection of the panel was determined for four compressive forces corresponding to 25%, 50%, 75% and 100% of the maximum force F_max_ (Figure 3).

At relatively small values of compressive force (below 2000 N) there was local loss of stability of the plate between the stringers in both variants (riveted and welded with pitch spacing *s* = 29.5 mm).

This phenomenon is manifested by the buckling of the plate in the form of three half-waves (Figure 4a). As the load increases, the buckling character of the plate gradually changes and transfers into the form of five half-waves, at a force value of about 7800 N for the welded variant, and 11 × 10^3^ N for the riveted variant (Figure 4b). In the final stage of the compression test, after a compressive force exceeding 30 × 10^3^ N, the panel deformation was characterized by the appearance of two dominant half-waves. Figure 5a shows the state of buckling of the welded plate immediately before reaching the value of critical force causing buckling of the stringers. These half-waves are also clearly visible on the inside of the panel between the stringers (Figure 5b). The buckling results revealed by the ARAMIS system are in accordance with the experimental investigations (Figure 6a,b).

It has been experimentally shown that the riveted and welded (*s* = 29.5 mm) panels exhibit elastic deformations in the range of compressive force between 0 and 42 × 10^3^ N (F_max_). The unloading of the panels does not reveal their plastic deformation. However, local plastic deformation of the plate was observed in the vicinity of some rivets and welds.

Table 2 shows the ultimate capacities of all specimens and models of both tests and numerical simulations with the quantitative differences in percentage. The differences between the values obtained from numerical modeling, and those obtained in the experiments, was about 11–14%. The numerical models in all the cases considered predicted higher ultimate capacities of the panels that were found in practice. This can be explained by the assumptions in the finite element model of the panel. The properties of all joints in numerical models of the stringer-stiffened panels were considered to be homogeneous. However, in the case of the experimentally fabricated RFSSW joints in particular, the properties of the weld do change along the weld radius and weld perimeter [13].

Figure 7a and Figure 8a show the numerically predicted deformation of the panel subjected to a compressive load of 0.25 F_max_. With both methods of joining, a buckling of the panel was observed, manifested by the existence of five half-waves between the stringers. In the case of a welded variant of the panel with a weld spacing *s* = 29.5 mm (Figure 7a), the sizes of half-waves oriented in both directions are similar. The maximum deviation of the panel surface from the plane of the undeformed sheet was approximately 1.19 mm. In the case of a panel with riveted longitudinal stringers (Figure 8a), the three half-waves that are oriented to the inside of the stiffened panel (Figure 2), where the stringers are placed, are smaller than two half-waves directed outside the stiffened panel. It should also be noted that the half-waves oriented inside the panel have an amplitude of approximately 1.36 mm, while those oriented outwards have an amplitude of approximately 1.17 mm.

For both variants considered, at a compressive load of 0.5 F_max_, a similar form of deformation states was noticed with two dominating half-waves, with an amplitude of approximately 2 mm, oriented towards the stiffened side (Figure 7b and Figure 8b).

The out-of-plane deformed shape of panels were measured along the mid line between stringers (Figure 9). Two dominant waves are clearly visible in the case of the deformation of riveted and welded panels (*s* = 29.5 mm) (Figure 10 and Figure 11). These waves were directed into the side of the panel where the stringers are located. The amount of deformation of the three other waves is smaller. After a load force reaches 0.5 F_max_, a clear localization of the deformation is visible in the case of the stiffened panel with welded stringers (*s* = 44.25 mm) (Figure 12). Despite precisely locating the plate deformation between stringers, a further increase of the axial load was observed. 

The diameter of the welds was three times larger than the rivet diameter. So the maximum value of the equivalent plastic stress in the welded panel (310 MPa) was smaller than in the case of the riveted stiffened panel which is exposed to equivalent plastic stress of 355 MPa (Figure 13a,b). At a panel load of 0.25 F_max_, the maximum equivalent stress value in the area of the plate’s joint with the stringers was 138.6 MPa, while in the riveted variant of panel the maximum equivalent stress values were 158 MPa. In riveted lap joints, loads are transferred by rivet shear and interference between faying sheets. The combined stresses produced by the hole and the uneven load transfer through the skin-stringer interface result in a complex 3D stress distribution [34].

Views of the deformed stiffened panel subjected to a compressive force of 0.75 F_max_ are shown in Figure 7c and Figure 8c. In the case of both joining techniques, the two half-waves were oriented outside the panel, which is in compliance with the results of the experiments. The amplitude of these two half-waves increases until the stringers buckle. 

The numerically predicted maximum values of equivalent plastic stress are similar to the ultimate tensile stress of the panel material and were 473 MPa for the welded variant, and 483 MPa for the riveted variant of the stiffened panel (Figure 14a,b). The areas in the vicinity of the connectors were the most affected by stress.

Figure 15a presents a view of the stiffened panel riveted with a spacing of *s* = 44.25 mm after the destruction of the weld. It is clear that there has been a local buckling of the sheet in the areas connecting the stringers with the base plate. Too high a spacing between welds caused a local reduction in panel stiffness (Figure 15b).

Figure 16a,b shows the distribution of equivalent plastic stress of the panel welded with a connector spacing of *s* = 44.25 mm. Local buckling was observed between welds in a welded panel with an increased weld spacing *s* = 44.25 mm. It can be noticed that due to these phenomena, the normal stresses dominate, which leads to the peeling of the panel components. The stiffened panel exhibits higher strength and stiffness in the case of operation in shear stress conditions. The load capacity of welds subjected to shear stresses is significantly greater than that of tear stresses. The maximum equivalent plastic stress in the vicinity of welds reaches a value of approximately 470 MPa.

The variant of a panel with increased weld spacing was experimentally analyzed because the weld diameter was three times larger than the diameter of the rivets. Therefore, the increased weld spacing could hypothetically provide appropriate strength for the structure. However, as was found for rivets, the increasing of the weld spacing caused a decrease in the stiffness of the stringer-stiffened panel.

## 4. Conclusions

This paper presents the results of experimental and numerical investigations of thin-walled stringer-stiffened panels under axial compression. Two methods of joining the skin and stringers of the stiffened panels were analyzed. The main conclusions drawn are as follows:The stiffened panel with welded stringers with a spacing of *s* = 29.5 mm exhibited an ultimate load similar to the variant with a riveted panel.The increase of weld spacing by 50% was the cause of local buckling of the plate between welds. So, the welds were more exposed to peel stress, which leads to weld failure before the compression force achieved maximum value.The number of half-waves and the amount of plate deflection depends on the load capacity and method of joining of the skin-stringer interface in the stiffened panels.Until the load level was below 0.75 F_max_, the numerically predicted maximum values of equivalent plastic stress were similar to the ultimate tensile stress of the panel material, and are equal to 473 MPa for the welded variant, and 483 MPa for the riveted variant of the stiffened panel.Too large a spacing between welds causes a local reduction in panel stiffness. Due to a local buckling of a panel, the normal stresses dominate, and lead to peeling of the panel components.

## Figures and Tables

**Figure 1 materials-12-01785-f001:**
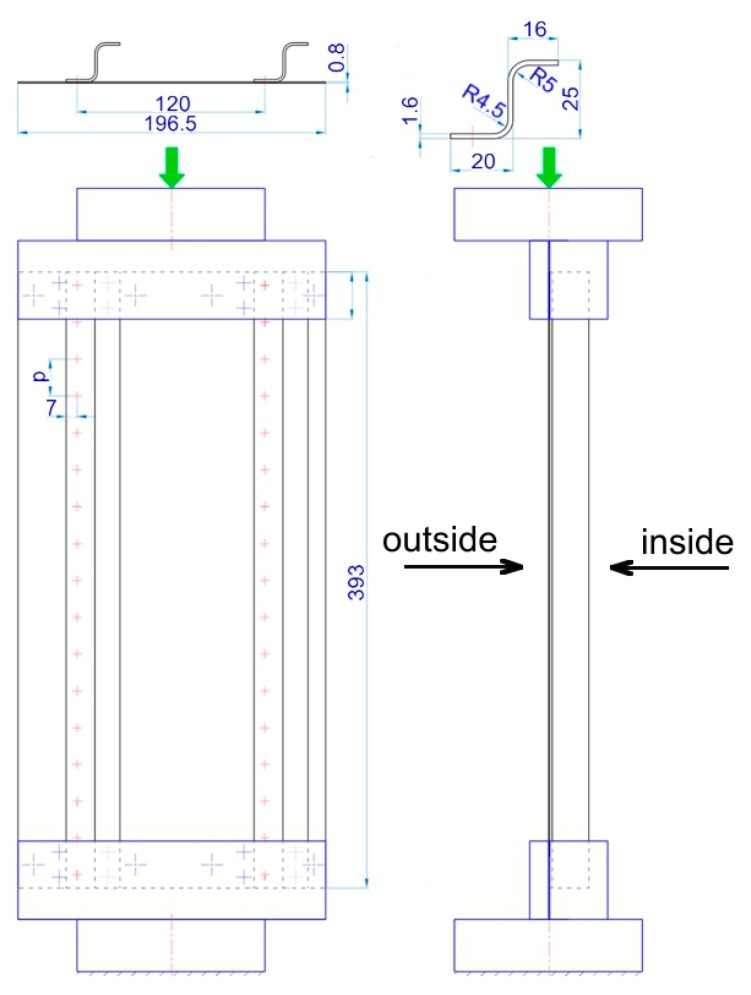
Dimensions of the panel and boundary conditions in the compression test.

**Figure 2 materials-12-01785-f002:**
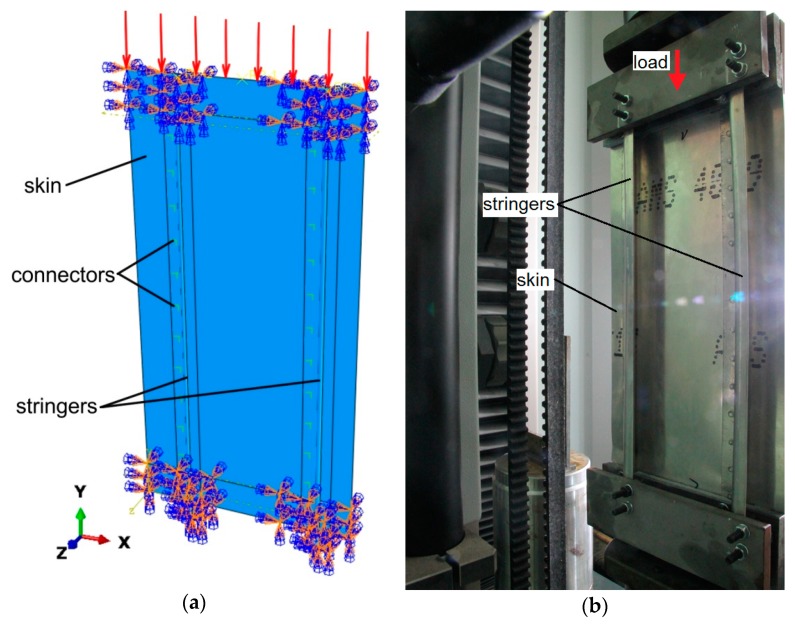
The boundary conditions on the model of panel compression analyzed (**a**) and a panel attached to the measuring stand (**b**).

**Figure 3 materials-12-01785-f003:**
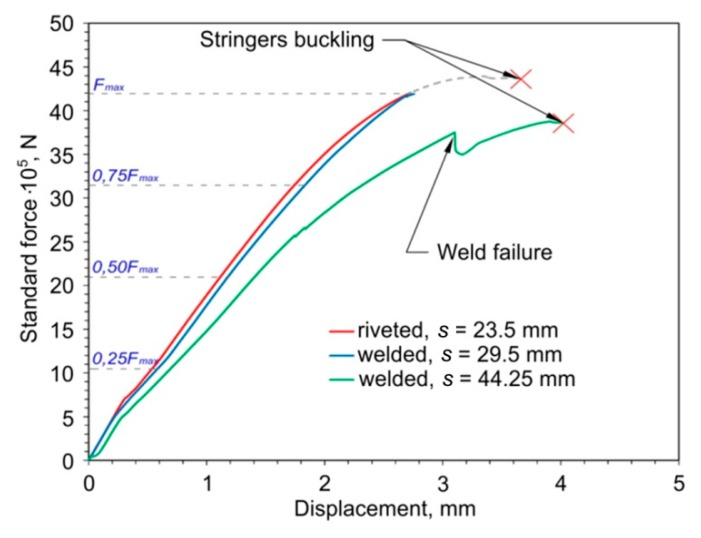
Force vs. displacement relationship for the variants of the panels analyzed.

**Figure 4 materials-12-01785-f004:**
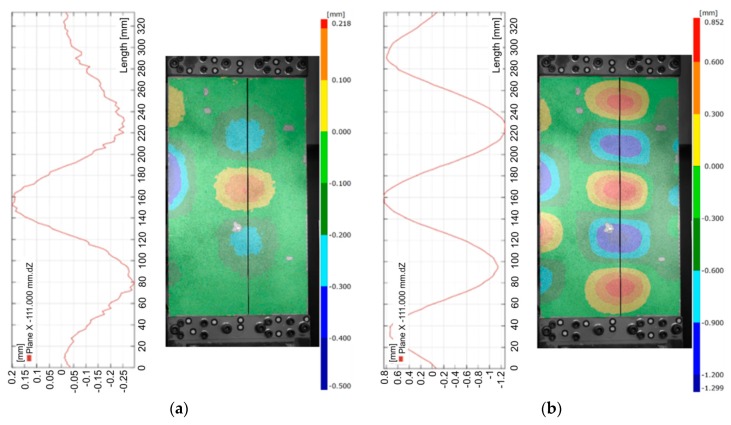
The deflection of riveted panels subjected to a compression load of 1800 N (**a**) and 9000 N (**b**) obtained using the three-dimensional digital image correlation (DIC) technique.

**Figure 5 materials-12-01785-f005:**
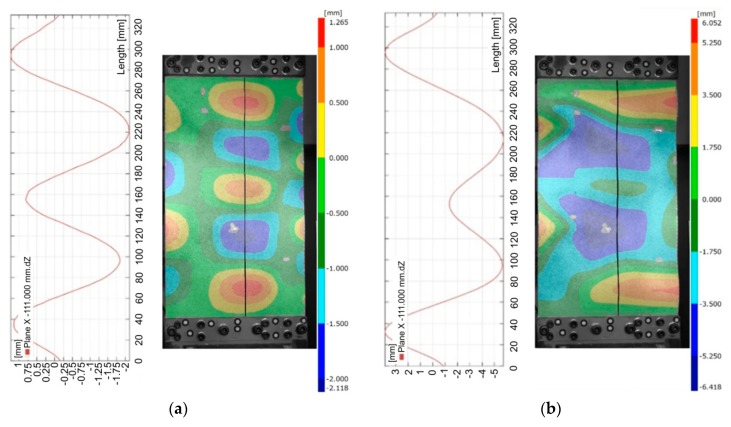
Deflection of riveted panels subjected to a compression load of 24 × 10^3^ N (**a**) and 43 × 10^3^ N (**b**) obtained using the three-dimensional DIC technique.

**Figure 6 materials-12-01785-f006:**
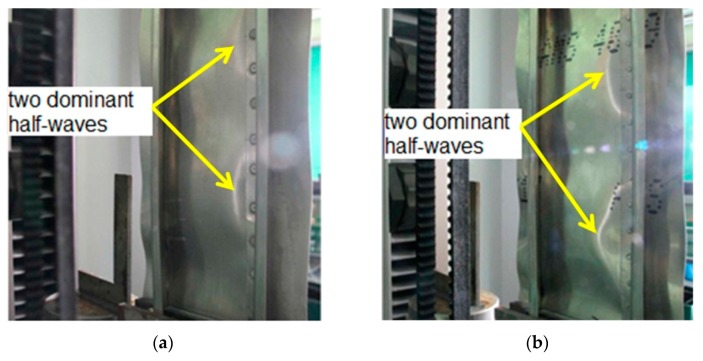
View of the postbuckling deformation of a panel subjected to a compression load of 4.2 × 10^3^ N: (**a**) Welded variant, *s* = 29.5 mm; (**b**) riveted variant, *s* = 23.5 mm.

**Figure 7 materials-12-01785-f007:**
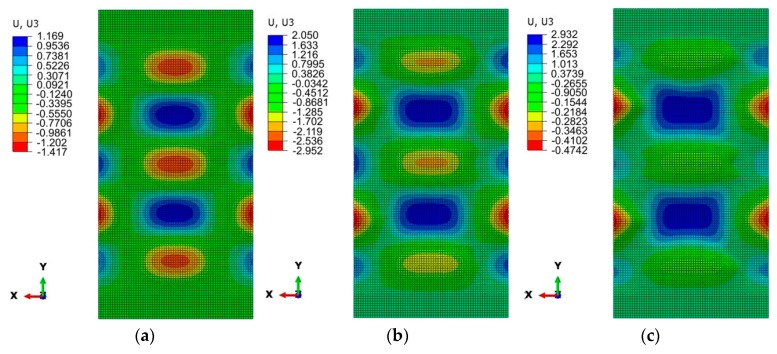
Deformation of a stiffened panel with welded stringers in a postcritical state (*s* = 29.5 mm) when subjected to a compressive force of: 0.25 F_max_ (**a**), 0.5 F_max_ (**b**) and 0.75 F_max_ (**c**).

**Figure 8 materials-12-01785-f008:**
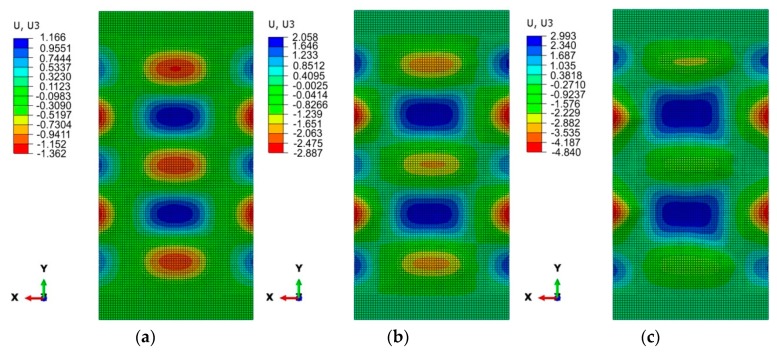
Deformation of a stiffened panel with riveted stringers in a postcritical state (*s* = 23.5 mm) when subjected to a compressive force of: 0.25 F_max_ (**a**), 0.5 F_max_ (**b**) and 0.75 F_max_ (**c**).

**Figure 9 materials-12-01785-f009:**
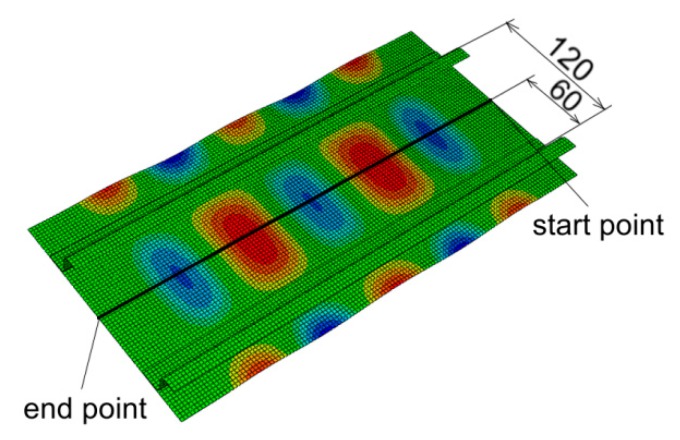
Path for measurement of out-of-plane deformation.

**Figure 10 materials-12-01785-f010:**
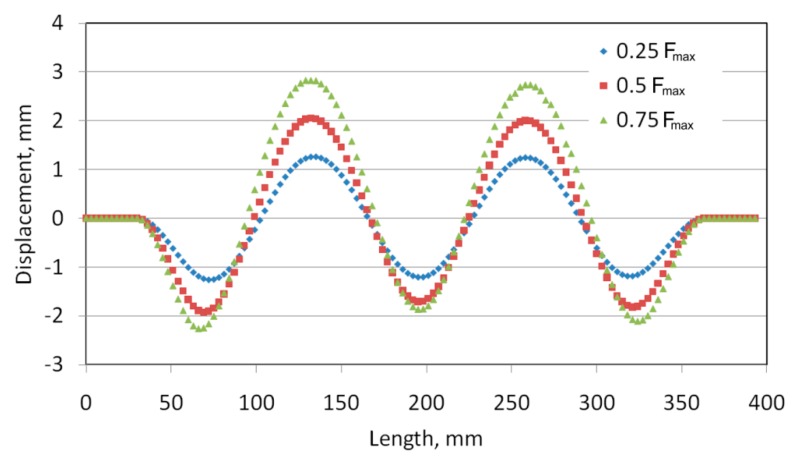
Out-of-plane deformation of the stiffened panel with riveted stringers (*s* = 23.5 mm).

**Figure 11 materials-12-01785-f011:**
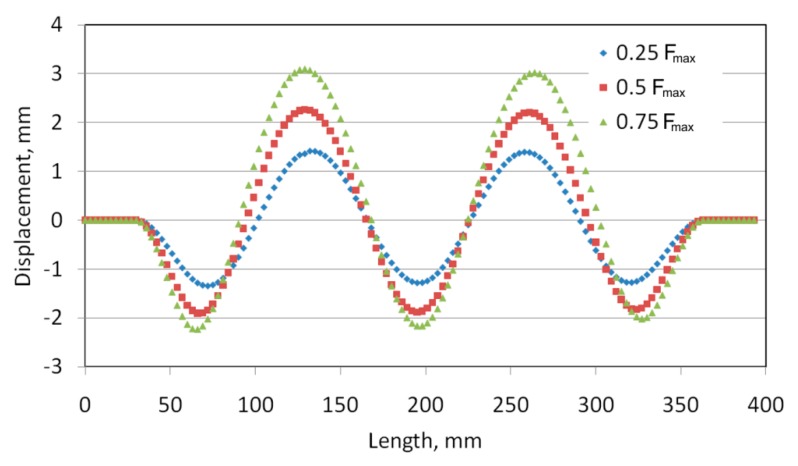
Out-of-plane deformation of the stiffened panel with welded stringers (*s* = 29.5 mm).

**Figure 12 materials-12-01785-f012:**
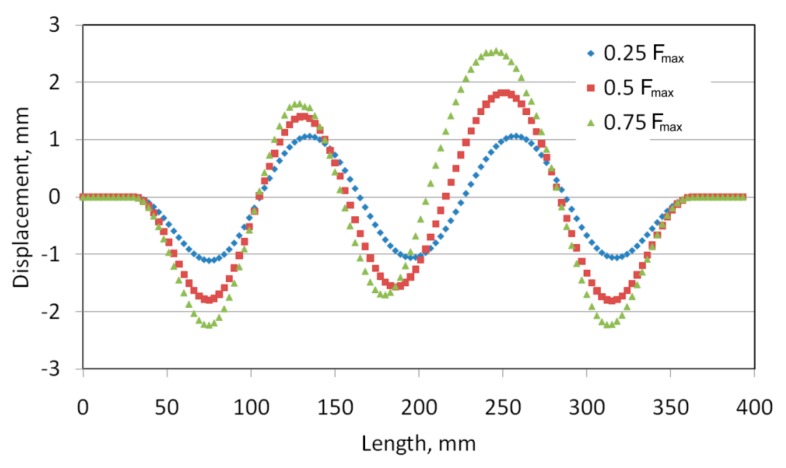
Out-of-plane deformation of the stiffened panel with welded stringers (*s* = 44.25 mm).

**Figure 13 materials-12-01785-f013:**
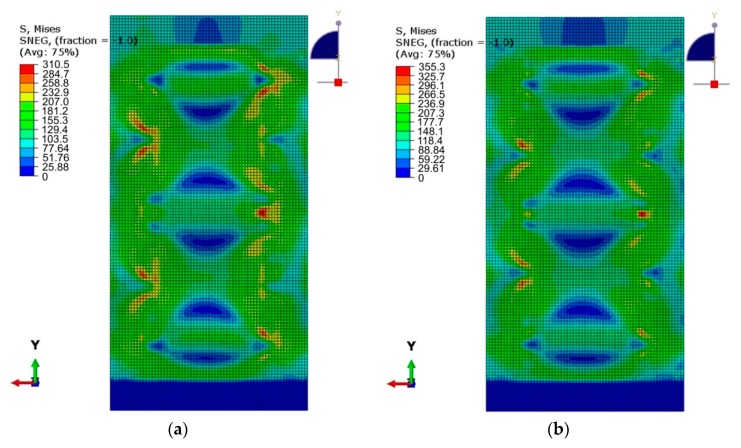
Comparison of the equivalent plastic stress on the outside of the stiffened panel subjected to a compressive force of 0.5 F_max_: (**a**) welded variant (*s* = 29.5 mm), (**b**) riveted variant (*s* = 23.5 mm).

**Figure 14 materials-12-01785-f014:**
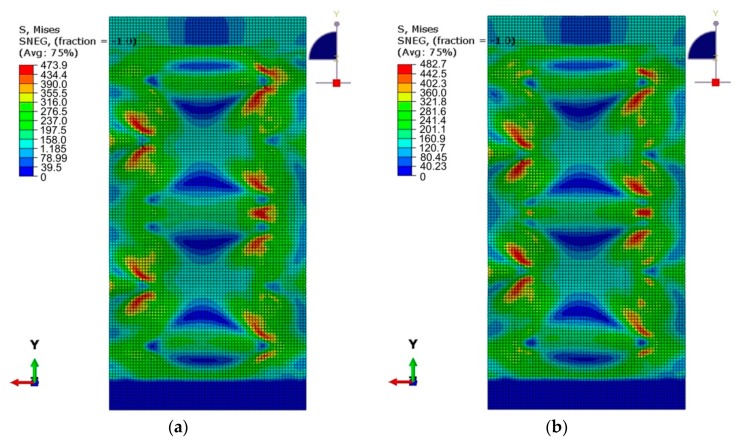
Comparison of equivalent plastic stress on the outside of a stiffened panel subjected to a compressive force of 0.75 F_max_: (**a**) welded variant (*s* = 29.5 mm), (**b**) riveted variant (*s* = 23.5 mm).

**Figure 15 materials-12-01785-f015:**
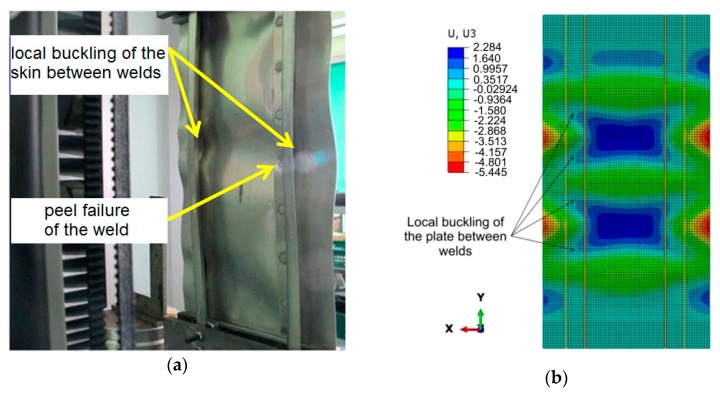
Deformation of a welded panel with a weld spacing of *s* = 44.25 mm subjected to a compressive force of 37.5 × 10^3^ N: (**a**) Experimental test, (**b**) numerical modeling.

**Figure 16 materials-12-01785-f016:**
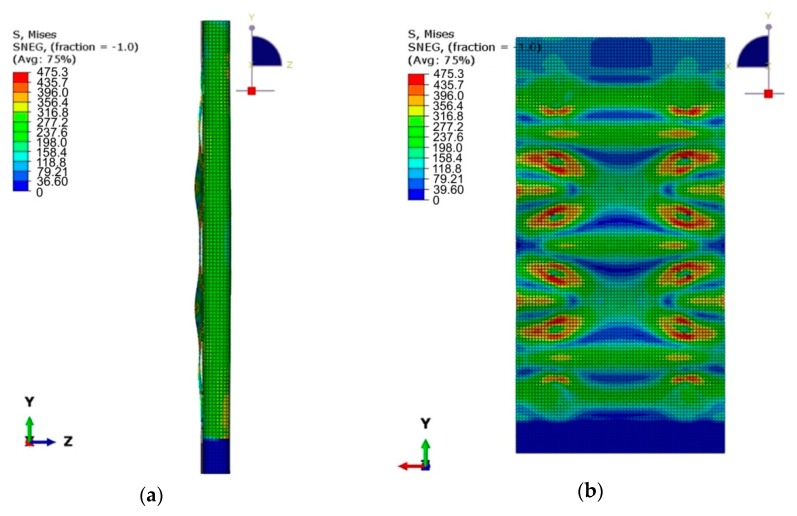
Local buckling of the panel between welds (**a**) and stress accumulation in the vicinity of the welds (**b**) for the welded stiffened panel with a weld spacing of *s* = 44.25 mm (compressive force of 0.75 F_max_).

**Table 1 materials-12-01785-t001:** The basic mechanical properties of Alclad 7075-T6 aluminum alloy.

Parameter	Ultimate Tensile Stress R_m_, MPa	Yield Stress R_p0.2_, MPa	Elongation A, %
Value	482.6	413.7	7

**Table 2 materials-12-01785-t002:** Comparison of experimental and numerically predicted load capacities of the panels.

Variant of Panel	Ultimate Capacity of Panel		Difference, %
	Experiment	Numerical Modeling	
riveted, *s* = 23.5	43 × 10^3^	48 × 10^3^	11.62
welded, *s* = 29.5	43 × 10^3^	49 × 10^3^	13.95
welded, *s* = 44.25	37 × 10^3^	42 × 10^3^	13.51

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
