# Peer review of "Experimental and Numerical Investigations of Thin-Walled Stringer-Stiffened Panels Welded with RFSSW Technology under Uniaxial Compression"

_materials, 2019, doi:10.3390/ma12111785_

Round 1
Reviewer 1 Report
· English needs some revision and proof reading that the authors need to undertake. E.g. “three-fold higher that rivet diameter”, “Until the load capacity reach of, “Due to a local buckling of a panel dominate” and so many others.
· Please add axial load-displacement graphs of the entire specimens.
· The curves of fig 4,5 are not clear enough.
· “Both experimental and numerical analyses are conducted”. Don’t use the same words both in abstract and introduction. You could paraphrase.
· Tabulate the ultimate capacities of all specimens and models of both tests and FE with the quantitative differences in percentage.
· You have used two types of connection that both require residual stresses. Using epoxy is mentioned for thin elements in doi.org/10.1016/j.tws.2014.08.023. Try to make short comparison.
· “weld was destroyed”: change the language.
· “p” should not represent the distance.
Author Response
English needs some revision and proof reading that the authors need to undertake. E.g. “three-fold higher that rivet diameter”, “Until the load capacity reach of, “Due to a local buckling of a panel dominate” and so many others.
English has been improved.
· Please add axial load-displacement graphs of the entire specimens.
Three panels were prepared for the riveted variant. The load-displacement characteristics in axial loading for these panels are very similar. So, we decided to prepare one panel for each spacing between welds that was considered. We added the comment that one panel has been tested for each configuration. So, Figure 3 consists of the results of all the experiments.
The curves of fig 4,5 are not clear enough.
Figures 4 and 5 have been corrected.
· “Both experimental and numerical analyses are conducted”. Don’t use the same words both in abstract and introduction. You could paraphrase.
Last paragraph in Introduction section has been rewritten.
· Tabulate the ultimate capacities of all specimens and models of both tests and FE with the quantitative differences in percentage.
Table 2 with description has been added.
· You have used two types of connection that both require residual stresses. Using epoxy is mentioned for thin elements in doi.org/10.1016/j.tws.2014.08.023. Try to make short comparison.
The section “Introduction” has been updated according to your comment.
· “weld was destroyed”: change the language.
Corrected.
· “p” should not represent the distance.
“p” has been replaced by “s”.
Reviewer 2 Report
This work is interesting and written in a professional way. The main novelty of this paper is in specific technical application and a contrast of the FEM results with experimentation.
Nevertheless, the Authors are encouraged to attach some matrix equations being solved by ABAQUS and a constitutive law applied for the material inserted into the stiffness matrix.
Some fundamental works for computational buckling analysis would be demanded.
A problem of calibration of the computer model with experimental works in the area of metal structures have been considered before, cf. J. Szafran. K. Juszczyk, M. Kamiński, Experiment-based reliability analysis of structural joints in a steel lattice tower. Journal of Constructional Steel Research 154: 278-292, 2019.
The paper should be published after these modifications.
Author Response
This work is interesting and written in a professional way. The main novelty of this paper is in specific technical application and a contrast of the FEM results with experimentation.
Nevertheless, the Authors are encouraged to attach some matrix equations being solved by ABAQUS and a constitutive law applied for the material inserted into the stiffness matrix.
Section 2.5. Numerical modeling has been updated.
Some fundamental works for computational buckling analysis would be demanded.
A problem of calibration of the computer model with experimental works in the area of metal structures have been considered before, cf. J. Szafran. K. Juszczyk, M. Kamiński, Experiment-based reliability analysis of structural joints in a steel lattice tower. Journal of Constructional Steel Research 154: 278-292, 2019.
The section “Introduction” has been updated according to your comment.
The paper should be published after these modifications.
Reviewer 3 Report
The work presents interesting experimental and numerical analyses about a thin-walled stringer-stiffened panel under axial compression. The topic is interesting but the manuscript could much better describe the work and the related results. Besides the initial part of the introduction, English does not match at all the standards of a scientific article. The reviewer encourages to strongly revise the overall English and to rephrase several parts.
At page 2, when stating: “Many papers are devoted to the strength analysis of the FSW fabricated joints”, it could worth refer to the following article in which cracks induced by FSW in stiffened panel were analysed:
· R. Sepe, E. Armentani, P. di Lascio, R. Citarella, Crack growth behaviour of welded stiffened panel, Procedia Eng, 109 (2015), pp. 473-483.
Moreover, the following article reported investigations of FSW welding parameters on the fatigue crack-growth behaviour of joints:
· R. Citarella, P. Carlone, R. Sepe, M. Lepore, DBEM crack propagation in friction stir welded aluminum joints, Adv Eng Softw, 101 (2016), pp. 50-59.
A picture of the experimental tests should be added to Fig. 3 at page 5.
There are two figures numbered as Fig. 3. Please double-check.
More information and/or references should be added when describing the material ABAQUS implementation.
Last sentence of page 6 states that: “The deflection of the panel was determined for four compressive forces 1800 N, 9000 N, 24·103 N and 43·103 N corresponded to the 25%, 50%, 75 % and 100% of the maximum force Fmax (Fig. 3).”. Please double check the figures and/or rephrase the sentence.
Some plots showing the out-of-plane deformed shape obtained numerically could be added to let the reader better understand the phenomena.
English must be improved before publishing.
Author Response
The work presents interesting experimental and numerical analyses about a thin-walled stringer-stiffened panel under axial compression. The topic is interesting but the manuscript could much better describe the work and the related results. Besides the initial part of the introduction, English does not match at all the standards of a scientific article. The reviewer encourages to strongly revise the overall English and to rephrase several parts.
English has been improved.
At page 2, when stating: “Many papers are devoted to the strength analysis of the FSW fabricated joints”, it could worth refer to the following article in which cracks induced by FSW in stiffened panel were analysed:
· R. Sepe, E. Armentani, P. di Lascio, R. Citarella, Crack growth behaviour of welded stiffened panel, Procedia Eng, 109 (2015), pp. 473-483.
Moreover, the following article reported investigations of FSW welding parameters on the fatigue crack-growth behaviour of joints:
· R. Citarella, P. Carlone, R. Sepe, M. Lepore, DBEM crack propagation in friction stir welded aluminum joints, Adv Eng Softw, 101 (2016), pp. 50-59.
The analyses conducted in the cited papers have been referenced.
A picture of the experimental tests should be added to Fig. 3 at page 5.
Figure 2b has been added.
There are two figures numbered as Fig. 3. Please double-check.
Corrected.
More information and/or references should be added when describing the material ABAQUS implementation.
Section 2.5. Numerical modeling has been updated.
Last sentence of page 6 states that: “The deflection of the panel was determined for four compressive forces 1800 N, 9000 N, 24·103 N and 43·103 N corresponded to the 25%, 50%, 75 % and 100% of the maximum force Fmax (Fig. 3).”. Please double check the figures and/or rephrase the sentence.
This sentence has been corrected.
Some plots showing the out-of-plane deformed shape obtained numerically could be added to let the reader better understand the phenomena.
Plots have been added.
English must be improved before publishing.
English has been improved.
Round 2
Reviewer 1 Report
The paper is now acceptable for publication.
Reviewer 2 Report
The Authors have extended and completed their work satisfactorily and now the article can be recommended for publication in this journal.